# Genome-Wide Association Study Identifies the Crucial Candidate Genes for Teat Number in Crossbred Commercial Pigs

**DOI:** 10.3390/ani13111880

**Published:** 2023-06-05

**Authors:** Lijuan Yang, Xuehua Li, Zhanwei Zhuang, Shenping Zhou, Jie Wu, Cineng Xu, Donglin Ruan, Yibin Qiu, Hua Zhao, Enqin Zheng, Gengyuan Cai, Zhenfang Wu, Jie Yang

**Affiliations:** 1College of Animal Science and National Engineering Research Center for Breeding Swine Industry, South China Agricultural University, Guangzhou 510642, China; ljyang@stu.scau.edu.cn (L.Y.); xuehuali1998@163.com (X.L.); zwzhuang@outlook.com (Z.Z.); shenpingzhou1109@163.com (S.Z.); wujiezi163@163.com (J.W.); cnxu@stu.scau.edu.cn (C.X.); ruandl@stu.scau.edu.cn (D.R.); 13422157044qyb@gmail.com (Y.Q.); eqzheng@scau.edu.cn (E.Z.); cgy0415@163.com (G.C.); 2National S&T Innovation Center for Modern Agricultural Industry, Guangzhou 510642, China; kczx_zh@163.com; 3Key Laboratory of South China Modern Biological Seed Industry, Ministry of Agriculture and Rural Affairs, Guangzhou 510642, China; 4Guangdong Provincial Key Laboratory of Agro-Animal Genomics and Molecular Breeding, South China Agricultural University, Guangzhou 510642, China; 5Yunfu Subcenter of Guangdong Laboratory for Lingnan Modern Agriculture, Yunfu 527400, China

**Keywords:** DLY pigs, GWAS, SNP, number of teats, *ABCD4*

## Abstract

**Simple Summary:**

Improving the reproductive performance of sows is a paramount goal in breeding programs. The number of teats is significantly associated with the reproductive performance of sows, with quantitative trait loci (QTL) related to porcine papilla number traits identified across almost all chromosomes, and with candidate genes proposed. In this study, we analyzed the total teat number traits of 1518 Duroc × (Landrace × Yorkshire) (DLY) pigs and performed genotyping using a 50K chip. By screening SNP loci and related genes, this study aimed to identify genetic factors affecting the number of teats in DLY pigs and to facilitate future breeding programs aimed at improving this trait.

**Abstract:**

The number of teats is a crucial reproductive trait with significant economic implications on maternal capacity and litter size. Consequently, improving this trait is essential to facilitate genetic selection for increased litter size. In this study, we performed a genome-wide association study (GWAS) of the number of teats in a three-way crossbred commercial Duroc × (Landrace × Yorkshire) (DLY) pig population comprising 1518 animals genotyped with the 50K BeadChip. Our analysis identified crucial quantitative trait loci (QTL) for the number of teats, containing the *ABCD4* and *VRTN* genes on porcine chromosome 7. Our results establish SNP variants of *ABCD4* and *VRTN* as new molecular markers for improving the number of teats in DLY pigs. Furthermore, the most significant noteworthy single nucleotide polymorphism (SNP) (7_97568284) was identified within the *ABCD4* gene, exhibiting a significant association with the total teat number traits. This SNP accounted for a substantial proportion of the genetic variance, explaining 6.64% of the observed variation. These findings reveal a novel gene on SSC7 for the number of teats and provide a deeper understanding of the genetic mechanisms underlying reproductive traits.

## 1. Introduction

In pig breeding, reproductive traits are paramount for economic productivity. A higher reproductive performance in the pig herd is directly related to increased production efficiency [1]. To promote the genetic progress of these traits, it is crucial to investigate related genes and understand their genetic mechanism. Teat number is an economically significant trait in breeding sows, as a larger litter size requires an adequate number of teats for lactation. The number of teats (NTE) is an essential indicator of the lactation capacity of sows. The absence of sufficient teats can adversely affect weight gain and mortality rates in piglets [2]. Furthermore, larger litters necessitate more teats for sows to provide necessary immunity and nutrients to piglets prior to weaning. QTL related to teat number have been identified on nearly all chromosomes, and candidate genes have been proposed [3]. As litter size increases, a sufficient number of teats is required to support piglets’ growth [4]. Heritability is an important factor to consider when selecting breeding animals, as it provides an estimate of how much of the trait variation is due to genetic factors. The trait of teat number in pigs is known to exhibit moderate heritability, which means that genetic factors are the major determinant of trait variation. This trait is known to vary between and within breeds, and it has been extensively studied in swine selection indices [5]. The variation in teat number between and within breeds has been suggested to be a result of both natural and human-driven artificial selection. Therefore, identifying quantitative trait loci (QTL) and candidate genes for teat number conformation traits at birth is scientifically and economically significant to provide markers for the genetic improvement of these traits in pigs. The attributes of porcine papillae exhibit a remarkable level of complexity and diversity, manifesting in four key aspects. Firstly, the number of papillae varies across different pig breeds and even among individuals within the same breed. Secondly, there exists a diverse range of papilla types, showcasing distinct morphological characteristics. Thirdly, the distribution of papillae across the abdomen displays regional specificity, with variations observed in different anatomical regions. Lastly, an intriguing feature is the asymmetrical nature of papilla distribution, highlighting differences between the left and right sides of the abdomen [4]. Mammary glands are a defining feature of the Mammalia class, yet the number and location of these glands vary greatly among species. Bilateral symmetry is a common feature of mammary glands, and variation in the number of functional mammary glands within a species is relatively low. However, a more comprehensive understanding of mammary gland development is required to fully exploit the genetic variability present in pigs [5].

Genome-wide association studies (GWAS) are a widely used approach for analyzing the associations between genotypes and phenotypes by testing differences in the allele frequency of single nucleotide polymorphisms (SNPs) between individuals [6,7,8,9]. Teat number represents a classical example of a polygenic quantitative trait. The utilization of high-density SNPs in GWAS presents a valuable avenue for unraveling the intricate genetic architecture underlying such complex traits [10,11]. Despite the identification of numerous QTL with broad effects on teat number, the underlying genetic architecture of this trait remains enigmatic. Previous studies have shown that the QTL for the NTE is a polygenic trait influenced by various QTL across almost all chromosomes of the pig genome [12,13]. The gene *VRTN* on SSC7 (*Sus scrofa* chromosome) has been identified as a possible candidate gene for NTE [11,12,13,14]. Although several QTL and candidate genes have been reported for NTE, the causal gene has yet to be conclusively confirmed. The regulation of porcine nipple number is believed to be influenced by multiple genetic factors, yet the precise identification of the major gene controlling this trait in pigs remains elusive. Moreover, there is a notable scarcity of molecular markers that possess breeding value for this specific trait. In summary, the regulatory mechanism underlying the porcine papilla number trait is intricate and multifaceted. Several SNPs associated with this trait have been identified on various chromosomes, leading to the identification of several significant candidate genes. Nonetheless, further research is necessary to unravel the precise molecular mechanisms governing this trait and to gain a more comprehensive understanding of its intricacies [10,11]. Thus, this study aimed to identify candidate genes for NTE using GWAS in a DLY pig population [15].

To achieve this goal, we conducted a GWAS to identify genomic regions that affect NTE traits in a three-way crossbred commercial DLY pig population. Furthermore, we initially investigated the expression levels of putative candidate genes at various developmental stages. Through our comprehensive investigation, we have discerned multiple statistically significant SNPs and compelling candidate genes associated with NTE. These findings not only expand the current knowledge of the genetic architecture of NTE, but also underscore the complex nature of this trait.

## 2. Materials and Methods

### 2.1. Ethics Statement

All animals used in the current study were managed in accordance with the guidelines of South China Agricultural University, Guangzhou, China, with approval number 2018F098. The experimental animals were obtained from the breeding pig farm Guangdong Wen’s Foodstuffs Group Co., Ltd., located in Guangdong, China.

### 2.2. Animal Samples and Genotyping and Quality Control

The experimental animals were sourced from a three-way crossbred commercial pig population, which involved the breeding of 84 Duroc boars with 397 Landrace × Yorkshire sows to produce a large-scale offspring. These pigs were subjected to uniform feeding conditions and were raised in farms owned by Wen’s Foodstuffs Group Co., Ltd. (Guangdong, China). Following the fattening period, 1518 individuals were humanely slaughtered for phenotype recording. Teat numbers were collected by separately counting the left and right teats after birth, and teat number calculation involved the summation of both sides. Genomic DNA was extracted from ear tissue samples, using the standard phenol/chloroform method, quantified, and diluted to 50 ng/μL. The 1518 DLY pigs were genotyped using the GeneSeek Porcine 50K BeadChip (Neogen, Lincoln, NE, USA), which contained 50,703 SNPs. Data quality control (QC) was performed using PLINK v1.07 [16]. Individuals with call rates of less than 95%, SNPs with call rates of less than 99%, and those with minor allele frequencies of less than 0.01 were excluded. SNPs that failed the Hardy–Weinberg equilibrium test (*p* < 10^−6^), and those unmapped or located on sex chromosomes were also excluded. Following QC, 28,065 SNPs for 1518 pigs remained for further analyses.

### 2.3. Population Structure

Principal component analysis (PCA) was performed using the qualified SNPs to investigate the population structure of the DLY pigs. PCA analysis was conducted using GCTA software (version 1.93.2 beta).

### 2.4. Genome-Wide Association Study

The univariate mixed linear model implemented in GEMMA software (version 0.98.5) [17] was used to perform GWAS on the number of teats phenotype using SNPs that passed the quality control filters. The first five principal components calculated by the GCTA tool [18] were included as covariates in the association analysis model. The model is expressed as follows:y = Wα + Xβ + u + ε
where y is the vector of teat number; W is the incidence matrix of covariates (fixed effects), including sex and the top five eigenvectors of PCA; α is a vector of the corresponding coefficients, including the intercept; X is the vector of all marker genotypes; β  represents the corresponding effect of marker size; u refers to an n × 1 vector of random effects, and ε is a vector of errors. The Bonferroni correction represents a highly stringent threshold for statistical significance. In this study, we employed the false discovery rate (FDR) procedure to determine the threshold *p*-values for single-locus GWAS [19]. Specifically, an FDR of 0.01 was used and the threshold *p*-value was defined as *p* = *FDR* × *N/M*, where N represents the number of SNPs with *p*-values < 0.01 in the GWAS results and M refers to the total number of SNPs qualified from the populations.

To estimate the phenotypic variance explained by genome-wide SNPs and the proportion of phenotypic variance explained by each significant SNP, we implemented a model using GCTA software [17]. The model is represented as follows:$$y\;=X\beta +g+\varepsilon \;\mathrm{with}\mathrm{var}\;(y)=A_{g\;}\sigma_{g}^{2}+I\;\sigma_{\varepsilon }^{2}\;$$
where y is the vector of teat number; β is the vector including fixed effects; X is an incidence matrix for β; g is the vector of the aggregate effects of all the qualified 50K SNPs for the pigs within one population; Ι is the identity matrix; $A_{g}$ is the genomic relatedness matrix estimated by these SNPs; $\;\sigma_{g}^{2}$ is the additive genetic variance captured by either the genome-wide SNPs or the selected SNPs; and $\;\sigma_{\varepsilon }^{2}$ is the residual variance.

### 2.5. Functional Candidate Genes Search

The potential functional genes were identified using the Ensemble Sus scrofa11.1 reference genome version (http://ensemble.org/Sus_scrofa/Info/Index, accessed 20 September 2022). Genes located nearest to significant SNPs were listed in the Tables. We used GeneCards, MGI, and NCBI public network databases to query the function of these genes. Additionally, we conducted a thorough literature review to gather information on the association between all candidate genes nearest to SNPs and then analyzed the NTE traits.

## 3. Results

### 3.1. Phenotype Statistic

The phenotypic statistics of NTE in DLY pigs are presented in Figure 1 for this study (Appendix A displays the specific phenotypic data for NTE in DLY pigs). The total number of teats ranged between 10 and 18, with a mean of 14.

### 3.2. Genome-Wide Association Studies

After inspecting the 50K SNP information of the DLY pigs, we conducted GWAS analysis on traits related to teat number. This analysis identified seven significant SNPs located on SSC7, SSC8, and SSC9 (Figure 2; Table 1). The seven SNPs reached the threshold (*p* = 1.19 × 10^−4^) at an FDR = 0.01. Based on the figure, it can be seen that the primary sites of correlation between the population and teat number phenotype are situated on SSC7, indicating that this site mainly influences the phenotypic variation of pig teat number. Notably, we detected one consistent QTL on SSC7 for teat number, with the top SNP demonstrating the highest phenotypic variance of teat number in this QTL, at 6.64%. Of the four SNPs located on SSC7, three were located within the *ABCD4* gene. *ABCD4* is a protein-coding gene belonging to the ATP-binding cassette transporter superfamily, and it participates in the intracellular processing of vitamin B12 transport [20]. It is thought that *ABCD4* plays a role in the intracellular processing of vitamin B12 transport, and mutations in the ATPase domain of this protein have been demonstrated to modify intracellular vitamin B12 trafficking [21]. Vitamin B12 acts as a necessary cofactor in methionine synthase, and low vitamin B12 levels, particularly during development, have been linked to elevated incidences of neural tube defects in humans [22,23].The mutations in *ABCD4* could potentially alter the expression or function of this gene. Further analyses could help to better characterize the variants in this region and identify important functional mutations. Therefore, *ABCD4* should be regarded as a critical candidate gene for teat number. Only one SNP on SSC7 was located within the *VRTN* gene. *VRTN* is a novel DNA-binding transcription factor that exclusively localizes in the nucleus, binds to DNA on a genome-wide scale, and regulates the transcription of a set of genes harboring *VRTN* binding motifs [24]. The functional role and underlying molecular mechanisms of certain candidate genes associated with the porcine papilla number trait still require further investigation. While these genes have shown potential significance in relation to the trait, additional studies are needed to elucidate their specific functions and the molecular pathways through which they influence papilla development [6].

## 4. Discussion

This study utilized a 50K SNP chip to identify SNP loci and associated genes responsible for the number of teats in DLY pigs. In addition, the QTL regions in the current study were significantly reduced in length relative to previous investigations. In this study, the genetic mechanisms governing mammary gland development in pigs were investigated through a GWAS on various measures of teat number in a composite population of commercial pigs. The study aimed to identify candidate genes that could influence mammary gland development and enhance lactation capacity through selective breeding. Individual counts of teats on each side were collected to evaluate bilateral symmetry and determine the most effective measure for selection, which was found to be the total number of teats. The study used advanced genetic analysis methods to identify SNP loci and related genes that influence teat number traits in DLY pigs. Previous studies have identified QTL related to porcine teat number traits on chromosomes and proposed candidate genes such as *VRTN*. Building on this prior research, the present study provides a more detailed understanding of the genetic factors that affect teat number traits in pigs. The results of this study may have implications for pig breeding programs, as they provide insights into the genetic basis of reproductive performance in DLY pigs. By identifying SNP loci and related genes that influence teat number, this study may help breeders select animals with better reproductive performance and ultimately improve the overall efficiency and profitability of pig farming.

In this study, we conducted a GWAS for NTE in a population of 1518 DLY pigs and identified a classic QTL on SSC7 that affects teat number. *ABCD4* and *VRTN* are two related genes which are likely responsible for the variation observed among individuals. Moreover, a most significant noteworthy SNP (7_97568284) was identified within the *ABCD4* gene, demonstrating a significant correlation with the total number of teats. The SNP explained the 8.68% phenotypic variance of teat number in the DLY pigs. SNP (7_97568284) was associated with TNE. This SNP is located in the 3′ UTR of the *ABCD4* gene. *ABCD4* has been reported to participate in the intracellular processing of vitamin B12 transport, and alterations in the ATPase domain of this protein have been shown to modulate intracellular vitamin B12 trafficking [20]. SNP (7_97617907) was associated with TNE. This SNP is located in the intron of the *VRTN* gene. *VRTN* is an intriguing DNA-binding transcription factor characterized by its exclusive nuclear localization. It exhibits genome-wide DNA binding capacity and effectively governs the transcriptional activity of a distinct set of genes that contain *VRTN* binding motifs [24]. Our results provide valuable new insights into the genetic architecture of TNE, providing novel markers for genetic improvement of teat number in DLY and related pig breeds. Despite the mapping of numerous QTL for complex traits in domesticated animals, only a limited number of mutations have been identified so far [6,7,8,9]. With the advancements in marker density, population size, and analytical methodologies, significant progress has been achieved in identifying and characterizing QTL associated with porcine teats number. However, it is worth noting that the majority of these studies have focused on Western commercial pig populations and purebred groups, while the exploration of Chinese local pig populations and their specific breeding strains remains relatively limited. Advancements in resequencing technology and cost reductions have led to remarkable progress in the genetic analysis of complex traits in livestock and poultry using resequencing data. In the future, integrating whole-genome resequencing data with detailed genomic annotation information, transcriptomics, and proteomics could enable precise localization of the major effect of QTL for porcine nipple number, narrow the confidence interval, and identify the causal gene.

The GWAS analysis revealed additional QTL associated with NTE, which are likely attributed to genetic variations occurring later in the developmental cascade of mammary gland formation [6,7,8,9]. Notably, distinct genetic factors have been identified to regulate specific pairs of mammary glands, including variations between left and right counterparts. Not only do these pairs exhibit divergent temporal appearances, but they also demonstrate differential molecular requirements and morphogenetic programs, emphasizing the inherent complexity of mammary gland development [24].

So far, it is evident that a dominant QTL on SSC7 significantly impacts the NTE. However, the causal genes and specific mutation sites in this QTL remain unidentified. To identify the function of the gene in the dominant QTL for NTE, we combined the findings of this study with previous reports, detecting the *ABCD4* gene as a strong candidate gene in the QTL responsible for the significant effect on NTE in SSC7. While the presence of mammary glands is a defining character of species in the class Mammalia [25], location and number of mammary glands across species are quite variable [26]. Mammary glands typically exhibit bilateral symmetry, and the variation in the number of functional mammary glands within a species is relatively low [27]. Mammary gland organogenesis begins during embryogenesis and continues throughout adulthood, with functional development and differentiation occurring during puberty and pregnancy. The study provides a foundation for further analysis of the causal genes involved in teat number. *ABCD4* is a member of the ATP-binding (ABC) transporter family responsible for processing vitamin B12 in cells. A previous study reported differential expression of *VRTN* in pig embryos at 17.5 dpc [24]. The formation and development of mammalian papillae and mammary glands exhibit fundamental similarities during the embryonic stage. This process can be divided into four distinct stages. In the pig’s embryonic stage, approximately 23 days after conception, the mammary line emerges as a result of the alignment of columnar and multi-level ectodermal cells on both sides of the abdomen. Within 24 to 36 h following the establishment of the mammary line, which occurs at approximately 26 days of embryonic development, the mammary line undergoes specialization, becoming visibly distinct. This critical developmental stage plays a pivotal role in determining the number of porcine papillae. The mammary gland substrate undergoes amplification and invagination into the underlying stromal layer, forming glandular buds. The epithelial cells within the glandular bud differentiate into the papillary skin, while the epithelial cells located below the glandular bud extend branching structures into the adipose plate, forming primitive mammary ducts and establishing primary mammary glands. At this stage, the development of the nipple and mammary gland tissue in the embryonic period is largely complete. Subsequently, these tissues will undergo further maturation during puberty and pregnancy after birth, ultimately culminating in the formation of functional lactating tissues and organs [26,27].

With the discovery of a repertoire of potential single nucleotide variant (SNV) genes derived from the population, which could potentially impact porcine populations, the clear trajectory for future investigation lies in identifying pigs harboring diverse SNV genotypes within or in close proximity to the *ABCD4* and *VRTN* genes. Subsequently, it is imperative to analyze the potential effects of these variants on the expression or regulatory patterns of host loci. This line of research will shed light on the functional implications and molecular mechanisms associated with these genetic variations, providing valuable insights into their influence on porcine traits and population dynamics [28]. The genes responsible for influencing the number of nipples in mammals, specifically causal or major genes, can exert their influence through their involvement in mammary gland development or by participating in related pathways that regulate the intricate process of mammary gland formation.

It is interesting to note that the candidate genes *VRTN*, which have been shown to influence the number of vertebrae in pigs, are also consistent with the candidate genes that are known to influence the number of nipples in pigs. This suggests that there may be a common genetic mechanism underlying the development of these two traits in pigs [24,28]. The identification of common candidate genes for multiple traits is not uncommon in genetics research and can provide valuable insights into the genetic architecture of complex traits. In mammals, the development of mammary gland complexes occurs along the mammary line, which extends bilaterally from the axilla to the inguin along the spine. At specific points along this line, mammary glands develop in pairs, and these points are determined by the underlying development of a vertebra. The formation of vertebrae arises from the somites, while the initiation of mammary gland formation is influenced by factors in the dermal mesenchyme, which also originates from the somites. Moreover, each individual mammary gland is believed to be governed by specific genetic components that dictate whether its development will be initiated and sustained [29,30,31,32].

The knowledge gained from these findings can potentially be used to improve pig breeding programs and increase efficiency in pig farming. It will be important for future research to further investigate the relationship between these candidate genes and the development of both vertebrae and nipples in pigs. This will help to provide a more comprehensive understanding of the genetic factors that influence these important traits and may lead to the development of new breeding strategies to improve the reproductive performance of pigs. Unraveling and validating causative mutations associated with oligogenic or polygenic traits, particularly non-coding variants, presents a formidable challenge in the absence of functional laboratory investigations. The task of identifying and confirming such mutations remains exceedingly difficult [33,34,35,36].

## 5. Conclusions

In summary, the number of teats in DLY pigs was analyzed using a 50K SNP chip, followed by GWAS analysis for the number of teats. Significant signal loci that affect the number of teats were located on chromosome 7, with the *VRTN* and *ABCD4* genes speculatively linked to the number of teats in this population. Our results provided novel insights into the genetic basis of porcine NTE. The study further supports the role of the *VRTN* and *ABCD4* gene regions in influencing the variability in the number of teats recorded in the DLY pig population. The next area of research will be to increase the marker density to search for the mutation sites of these genes and regions. Our findings provide a reference for understanding the genetic mechanisms and breeding improvement of the number of teat traits.

## Figures and Tables

**Figure 1 animals-13-01880-f001:**
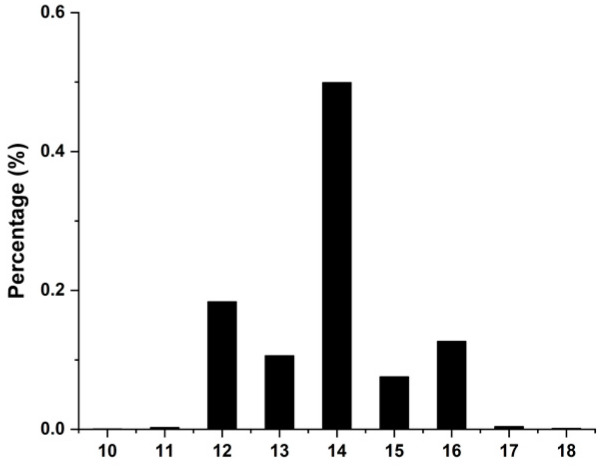
Phenotypic distribution of the number of teats in DLY pig populations.

**Figure 2 animals-13-01880-f002:**
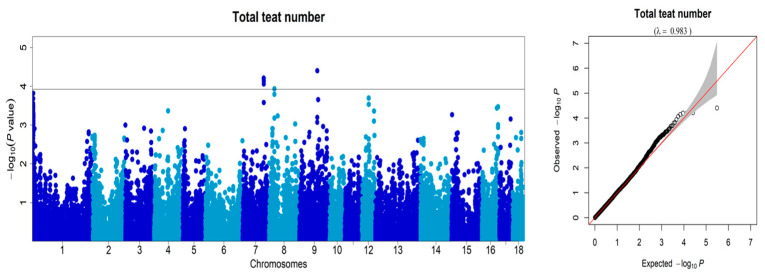
Manhattan plots and quantile-quantile (Q-Q) plots were generated to visualize the results of the GWAS for the number of teats traits in the DLY pig population. The X-axis represents the chromosome, and the Y-axis represents the −Log10 (*p*-value) of each SNP.

**Table 1 animals-13-01880-t001:** The seven SNPs significantly associated with the NTE in the DLY pig population were identified.

chr	SNP ID	Position (bp)	*p*-Value	Gene	Region
9	ALGA0054033	82,853,264	3.94 × 10^−5^	/	/
7	Affx-114687136	97,568,284	6.12 × 10^−5^	*ABCD4*	3′ UTR
7	Affx-115258151	97,595,573	6.29 × 10^−5^	*ABCD4*	5′ UTR
7	Affx-114892585	97,575,068	6.32 × 10^−5^	*ABCD4*	Intron 7-8
7	WU_10.2_7_103460706	97,617,907	7.26 × 10^−5^	*VRTN*	Intron 1-2
7	WU_10.2_7_103232787	97,584,287	8.67 × 10^−5^	*ABCD4*	Intron 1-2
8	ASGA0094767	24,709,455	1.16 × 10^−4^	/	/

## Data Availability

Data are contained within the article.

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
