# Peer review of "Genome-Wide Association Study Identifies the Crucial Candidate Genes for Teat Number in Crossbred Commercial Pigs"

_animals, 2023, doi:10.3390/ani13111880_

Round 1

Reviewer 1 Report

The authors present a research study leading to the identification of several SNPs that might be associated to teat number in pigs.

Overall, the article is well written and statistical analyses seem sound and well supported. However, I would like the authors to include more details regarding the following:

1. In section 2.2 of Methods the authors mention that after filtering, the remaining pigs analysed were 1518. But how many pigs were present before filtering? What was the input population originally? Also, could the authors state the percentage (or number) of each pig breed before and after filtering?

2. In section 2.6 of Methods the authors mention qPCR analyses being done on a set of animals to detect targeted gene expression. First, I would like a supplementary table to be included showing all the processing steps from raq Cq values up to final deltadeltaCq values. This would add clarity to the analyses done. Second, authors say that they report the primers used for qPCR capture in a supplementary table S1 but I do not have access to that table. Could you please provide it? Authors should specify primer length and annealing properties, as well as the software employed for their design. Third, authors need to report the number of animals on which the RNA extraction was performed, and their breeds.

3. In Results section 3.1, did the number of teats have any sort of variation across breeds? Since there is no clear number of animals reported to verify how powerful the analysis was, it is difficult to ascertain the reliability of these numbers.

4. The authors mention that the most significant SNPs are located "within" the ABCD4 gene, although in the Table 2 they mention that ABCD4 is the "nearest" gene. Inspection of the position of these SNPs in the Sscrofa11.1 genome reveal that some are indeed located "within" the ABCD4 gene, but others are in the upstream region. The authors should report where "within" or "near" these ABCD4 or VRTN genes are these SNPs located (introns, exons, 3UTR, 5UTR, upstream, downstream, promoter?).

5. The analysis reported in section 3.3 of the Results is somewhat irrelevant, although complementary to the analysis that should be reported instead. The authors there compare the levels of expression of several genes across different time points. For this analysis to be relevant, authors need to report the number of animals involved in each temporal group. Also, I do not understand the motivation of including additional genes other than VRTN and ABCD4. Are these also showing significant SNPs? Are these genes close to VRTN or ABCD4? If not, better remove them or clearly explain why you measured those additional genes. However, the main flaw of this analysis is that what authors should measure by qPCR is if the presence of any of the significant SNPs reported in Table 2 have any effect in the expression of VRTN or ABCD4 genes. Instead, authors seem to have performed the qPCR analyses on mice?? as they mention in the Discussion, although that is not mentioned in the Methods. If authors want to make any argument about the functional implication of these SNPs in the genes they are located in, they need to measure the allele-dependent expression pattern of those genes in mammary tissues from pigs that are carriers of such SNPs. The qPCR analyses are therefore somewhat irrelevant and useless, at least given the implication that the authors want to give to them. Could the authors perform a qPCR analysis on a subset of pigs carrying differential alleles for some of the SNPs reported and measure any possible change in the expression of ABCD4 and VRTN genes?

Reviewer 2 Report

The article is generally good, though I think it could be further improved through some minor suggestions below:

Line 44 The heritability of NTE (TTN) was 0.49, which should not be low heritability.

Line 61-62 Why is transcriptional expression analysis performed on selected candidate genes using mice instead of observing the expression in pigs?

Line 75-76 Why are both female and male pigs included in the trihybrid population used for genome-wide association analysis, and why not use the same sex? It is possible that candidate genes identified in this genome-wide analysis are caused by gender-related factors.

Line 83-84 Is setting MAF to 0.01 too small?

Line 146-147 How is the chromosome-wide threshold calculated? (If it is 0.1/SNP, these loci may not be so significant.)

Figure3 Does validating the results of genome-wide association analysis in mice make sense for pigs and does it suggest that ABCD4 is also related to mammary development in pigs?

Round 2

Reviewer 1 Report

I acknowledge the methodological clarifications included in this new revision. However, I keep fairly skeptical about the author's approach using qPCR in mice and several additional genes completely unrelated to the focus of the manuscript.

Following some more detailed points that need to be addressed before this manuscript is suited for publication:

1. In Table 2 (which should be actually named as Table 1), the authors have added where in the genes are the significant SNPs located, which I appreciate. However, authors keep the "Nearest gene" header in the fifth column. To be clear, a SNP within a gene cannot be near the gene, but inside the gene. Thereby, the fifth header must be changed. Please remove the "Nearest" word as it confused the reader.

2. Section 3.3 of the results has not been modified in essence. In my opinion, this section and all the related discussion should be removed from the manuscript, as it does not add any information relevant for pig biology and the association of the significant SNPs with gene expression dynamics. The addition of genes that do not appear in any other section is also completely detached from the main focus of the manuscript. Please remove the qPCR analyses and the mention to additional genes other than those appearing in Table 2. Any other reference in the Discussion to the qPCR results and additional genes that do not appear in Table 2 should be removed too. These results are confusing and have detrimental value to the overal content of the manuscript. I will not endorse the publication of this research unless authors address properly this point.

3. From the overall conclusions, the fact the ABCD4 is more expressed than VRTN has nothing to do with their involvement in the phenotype reported by the authors and GWAS analyses on SNPs within these genes supposedly associated with changes in teat number. The initial focus of the manuscript is based on how teat number is associated with SNVs located in certain genes. If authors wan to develop from this, the proper approach would be to measure gene expression change according to genotypes of the significant SNVs. Authors can use different developing stages, that would add value to the comparisons, mirroring their design in mice. However, the relevant change should be measured relative to genotype variation. This would add relevant information about the involvement of the SNVs in modulating teat number as a function of their effect in gene expression. Another interesting approach could be using an eGWAS approach using gene expression data from RNAseq or qPCR on pigs with divergent genotypes for these SNVs. Linking epression GWAS (eGWAS) with phenotype GWAS (teat number) would be a relevant addition to the manuscript. Unfortunately, authors seem to have opted for qPCR in mice, which is absolutely irrelevant, I am afraid.

Should the authors decide to properly address points 1 to 3, I will revise an additional version of the manuscript.

Overall English usage is correct. Some phrases are not grammatically correct, mostly the new additions. Please be careful and revise the grammatical usage and coherence of the proposed topics.

Round 3

Reviewer 1 Report

Many thanks for this new version of the manuscript, which has been substantially improved. I have however detected several minor issues that should be corrected before the manuscript is acceptable for publication.

1. In line 35 of the Abstract, you mention ABCD4 to be a causal gene affecting teat number. To be clear, what you found is a significant association between one SNP within ABCD4 gene and a certain trait (NTE as termed in the manuscript), but you did not provide additional proof that this SNP is actually causing the observed differences in teat number in your pig population. Therefore you should not, by any means, use the word "causal" anywhere in the manuscript when referring to the significant association found. This SNP might be, or might not be responsible for the observed variance. You found a correlation, but you did not provide additional proof of the causation. Please be careful in how you interpret your data in light of the results obtained.

2. The paragraph in lines 71-81 of the Introduction is quite badly written and hard to follow. Also, some assertions are not supported by any reference. I strongly suggest to rewrite or remove this part of the Introduction.

3. Please just use "Gene" and "Region" as the headers for the last two columns of the Table 1.

4. Lines 201 to 218 of the Discussion is somewhat a redundant repetition of the Introduction. Please remove this paragraph and/or integrate it properly in the Introduction section.

5. I would not suggest to use the initiator "To conclude" in line 227 of the Discussion, as it implies you have already discussed the topic, when you are actually starting to do it. Also, you refer to "other regions", which I do not know what that refers to. Probably this is pointing to the other genes used for qPCR that were removed? Honestly, lines 227 to 241 are quite badly structured. I suggest them to be removed or properly rewritten to follow a proper path through the Discussion. You should start with the main aim, why studies on genetic variation affecting teat number might or might not be properly explored, to then start discussing the possible implications of your findings regarding polymorphisms in ABCD4 and VRTN loci.

6. The phrase in lines 304-306 must be supported with a reference.

7. You mention the potential of using CRSIPR technology to create knockout mice lines without ABCD4 or VRTN gene activity to better elucidate their impact in teat number. Although this line of research is correct, there is a more approachable and certainly cheaper direction of your research to further substantiate your results in future papers. Once you have identified a series of candidate genes with SNVs segregating in your population and potentially affecting teat number (at least showing a significant association), the obvious direction to further your research line would be, as said previously, to identify pigs with divergent genotypes for SNVs in ABCD4 and VRTN genes, or nearby genes, and analyse the impact those variants might have in the expression or regulation patterns of the hosting loci. This can be achieved by simple targeted qPCR or by RNA-seq. You could even investigate whether the SNVs have any structural implications. Do they create any splicing alteration (for those that are intronic)? Do they remove or add any regulatory sites for TFs or microRNAs within the UTRs (for those located in 3/5 UTRs), for instance? There are several venues that are fairly easy to implement as a way to strengthen your GWAS findings prior to investigating knockout mice. If you want to propose alternative lines of research in your Discussion for future developments, the first approach should not be to change the species and propose an expensive technique such as CRISPR knockout mice. First keep on your species, and propose far less expensive molecular techniques that could add actual direct proof to your findings. Knockout mice, although possibly successful to prove that ABCD4 or VRTN expression is needed for teat development, would still be an indirect proof that would require applying additional techniques in pigs to assess the real biological causes of the changes observed in teat number, as your pig population is actually not a knockout of these genes, nor you should expect to explain the underling molecular mechanism of teat number development by knocking out two genes. This also ignores the putative ethical repercusions on animal welfare to be tested. I strongly encourage the authors to develop on this issue in the Discussion.

8. Supplementary Table 1, as far as I could see, still reports the primers used for qPCR, while Supplementary Table 2 and its content is actually the one referred to in the manuscript as Supplementary Table 1. Please remove your current Suppl Table 1 with the qPCR primers, and change the name of Suppl Table 2 to Suppl Table 1.

9. As said before, please be extra careful when linking your results to a "causal" effect. Please remember that GWAS analyses are only pointing towards a significant statistical association/correlation. No causal proof can be inferred from GWAS analyses alone. Should the authors decide to develop causal inferences for their significant associations, they would need to implement targeted molecular analyses on their pig population to elucidate the underlying effects of these SNVs on gene expression and regulation patterns explaining the observed linked phenotype.  

English is overall good, although some sections need a rewriting and reformatting to allow a fluent read and flow of the ideas and findings highlighted in the manuscript.
